# Post-lockdown SARS-CoV-2 nucleic acid screening in nearly ten million residents of Wuhan, China

Shiyi Cao[1,11], Yong Gan[1,11], Chao Wang[1,11], Max Bachmann[2], Shanbo Wei[3], Jie Gong[4], Yuchai Huang[1], Tiantian Wang[1], Liqing Li[5], Kai Lu[6], Heng Jiang[7,8], Yanhong Gong[1], Hongbin Xu[1], Xin Shen[1], Qingfeng Tian[9], Chuanzhu Lv[10 ✉], Fujian Song [2 ✉], Xiaoxv Yin[1 ✉] & Zuxun Lu [1 ✉]

Stringent COVID-19 control measures were imposed in Wuhan between January 23 and April 8, 2020. Estimates of the prevalence of infection following the release of restrictions could inform post-lockdown pandemic management. Here, we describe a city-wide SARS-CoV-2 nucleic acid screening programme between May 14 and June 1, 2020 in Wuhan. All city residents aged six years or older were eligible and 9,899,828 (92.9%) participated. No new symptomatic cases and 300 asymptomatic cases (detection rate 0.303/10,000, 95% CI 0.270–0.339/10,000) were identified. There were no positive tests amongst 1,174 close contacts of asymptomatic cases. 107 of 34,424 previously recovered COVID-19 patients tested positive again (re-positive rate 0.31%, 95% CI 0.423–0.574%). The prevalence of SARS-CoV-2 infection in Wuhan was therefore very low five to eight weeks after the end of lockdown.

[1] Department of Social Medicine and Health Management, School of Public Health, Tongji Medical College, Huazhong University of Science and Technology, Wuhan, Hubei, China. [2] Norwich Medical School, Faculty of Medicine and Health Science, University of East Anglia, Norwich, UK. [3] Wuhan Municipal Health Commission, Wuhan, Hubei, China. [4] Wuhan Centre for Clinical Laboratory, Wuhan, Hubei, China. [5] Department of Management Science and Engineering, School of Economics and Management, Jiangxi Science and Technology Normal University, Nanchang, Jiangxi, China. [6] Tongji Hospital, Huazhong University of Science and Technology, Wuhan, Hubei, China. [7] Centre for Alcohol Policy Research, School of Psychology and Public Health, La Trobe University, Melbourne, VIC, Australia. [8] Melbourne School of Population and Global Health, University of Melbourne, Melbourne, VIC, Australia. [9] School of Public Health, Zhengzhou University, Zhengzhou, Henan, China. [10] Department of Emergency, Hainan Clinical Research Centre for Acute and Critical Diseases, The Second Affiliated Hospital of Hainan Medical University, Haikou, Hainan, China. [11] These authors contributed equally: Shiyi Cao, Yong Gan, Chao Wang. ✉email: lyuchuanzhu@hainmc.edu.cn; Fujian.song@uea.ac.uk; yxx@hust.edu.cn; zuxunlu@yahoo.com

The Coronavirus Disease 2019 (COVID-19) was first reported in December 2019, and was classified as a pandemic by the World Health Organization on March 11, 2020[1]. Following strict lockdown measures, the COVID-19 epidemic was generally under control in China, and the whole country has progressed into a post-lockdown phase. In this phase, countries face new problems and challenges, including how to accurately assess the post-lockdown risk of the COVID-19 epidemic, how to avoid new waves of COVID-19 outbreaks, and how to facilitate the resumption of economy and normal social life. As the city most severely affected by COVID-19 in China, Wuhan had been under lockdown measures from January 23 until April 8, 2020. During the first 2 months after city's reopening, there were only a few sporadic COVID-19 cases in Wuhan (six newly confirmed cases from April 8 to May 10, 2020[2]). However, there was still concern about the risk of COVID-19 in Wuhan, which seriously affected the resumption of industrial production and social services, and hampered the normal lives of residents. In order to ascertain the current status of the COVID-19 epidemic, the city government of Wuhan carried out a comprehensive citywide nucleic acid screening of SARS-CoV-2 infection from May 14, 2020 to June 1, 2020.

The citywide screening of SARS-CoV-2 infection in Wuhan is a mass screening programme in post-lockdown settings, and provided invaluable experiences or lessons with international relevance as more countries and cities around the world entering the post-lockdown phase. In this study, we report the organisation process, detailed technical methods used, and results of this citywide nucleic acid screening.

## Results

There were 10,652,513 eligible people aged ≥6 years in Wuhan (94.1% of the total population). The nucleic acid screening was completed in 19 days (from May 14, 2020 to Jun 1, 2020), and tested a total of 9,899,828 persons from the 10,652,513 eligible people (participation rate, 92.9%). Of the 9899,828 participants, 9,865,404 had no previous diagnosis of COVID-19, and 34,424 were recovered COVID-19 patients.

The screening of the 9,865,404 participants without a history of COVID-19 found no newly confirmed COVID-19 cases, and identified 300 asymptomatic positive cases with a detection rate of 0.303 (95% CI 0.270–0.339)/10,000. The median age-stratified Ct-values of the asymptomatic cases were shown in Supplementary Table 1. Of the 300 asymptomatic positive cases, two cases came from one family and another two were from another family. There were no previously confirmed COVID-19 patients in these two families. A total of 1174 close contacts of the asymptomatic positive cases were traced, and they all tested negative for the COVID-19. There were 34,424 previously recovered COVID-19 cases who participated in the screening. Of the 34,424 participants with a history of COVID-19, 107 tested positive again, giving a repositive rate of 0.310% (95% CI 0.423–0.574%).

Virus cultures were negative for all asymptomatic positive and repositive cases, indicating no "viable virus" in positive cases detected in this study.

All asymptomatic positive cases, repositive cases and their close contacts were isolated for at least 2 weeks until the results of nucleic acid testing were negative. None of detected positive cases or their close contacts became symptomatic or newly confirmed with COVID-19 during the isolation period. In this screening programme, single and mixed testing was performed, respectively, for 76.7% and 23.3% of the collected samples. The asymptomatic positive rates were 0.321 (95% CI 0.282–0.364)/10,000 and 0.243 (95% CI 0.183–0.315)/10,000, respectively.

The 300 asymptomatic positive persons aged from 10 to 89 years, included 132 males (0.256/10,000) and 168 females (0.355/10,000). The asymptomatic positive rate was the lowest in children or adolescents aged 17 and below (0.124/10,000), and the highest among the elderly aged 60 years and above (0.442/10,000) (Table 1). The asymptomatic positive rate in females (0.355/10,000) was higher than that in males (0.256/10,000).

The asymptomatic positive cases were mainly domestic and unemployed residents (24.3%), retired older adults (21.3%), and public service workers (11.7%) (Fig. 1).

The asymptomatic positive rate in urban districts was on average 0.456/10,000, ranging from 0.317/10,000 in Hongshan to 0.807/10,000 in Wuchang district. A lower rate of asymptomatic positive cases was found in suburban districts (0.132/10,000), ranging from 0.047/10,000 in Xinzhou to 0.237/10,000 in Jiangan district (Fig. 2).

Among the 7280 residential communities in Wuhan, asymptomatic positive cases were identified in 265 (3.6%) communities (only one case detected in 246 communities), while no asymptomatic positive cases were found in other 96.4% communities.

Testing of antibody against SARS-CoV-2 virus was positive IgG (+) in 190 of the 300 asymptomatic cases, indicating that 63.3% (95% CI 57.6–68.8%) of asymptomatic positive cases were actually infected. The proportion of asymptomatic positive cases with both IgM (−) and IgG (−) was 36.7% (95% CI: 31.2–42.4%; $n = 110$), indicating the possibility of infection window or false positive results of the nucleic acid testing (Table 2).

Higher detection rates of asymptomatic infected persons were in Wuchang, Qingshan and Qiaokou districts, and the prevalence of previously confirmed COVID-19 cases were 68.243/10,000, 53.767/10,000, and 100.047/10,000, respectively, in the three districts. Figure 3 shows that districts with a high detection rate of asymptomatic positive persons generally had a high prevalence of confirmed COVID-19 cases ($r_s = 0.729$, $P = 0.002$).

## Discussion

The citywide nucleic acid screening of SARS-CoV-2 infection in Wuhan recruited nearly 10 million people, and found no newly confirmed cases with COVID-19. The detection rate of asymptomatic positive cases was very low, and there was no evidence of transmission from asymptomatic positive persons to traced close contacts. There were no asymptomatic positive cases in 96.4% of the residential communities.

Previous studies have shown that asymptomatic individuals infected with SARS-CoV-2 virus were infectious[3], and might subsequently become symptomatic[4]. Compared with symptomatic patients, asymptomatic infected persons generally have low quantity of viral loads and a short duration of viral shedding, which decrease the transmission risk of SARS-CoV-2[5]. In the present study, virus culture was carried out on samples from asymptomatic positive cases, and found no viable SARS-CoV-2 virus. All close contacts of the asymptomatic positive cases tested negative, indicating that the asymptomatic positive cases detected in this study were unlikely to be infectious.

There was a low repositive rate in recovered COVID-19 patients in Wuhan. Results of virus culturing and contract tracing found no evidence that repositive cases in recovered COVID-19 patients were infectious, which is consistent with evidence from other sources. A study in Korea found no confirmed COVID-19 cases by monitoring 790 contacts of 285 repositive cases[6]. The official surveillance of recovered COVID-19 patients in China also revealed no evidence on the infectiousness of repositive cases[7]. Considering the strong force of infection of COVID-19[8–10], it is expected that the number of confirmed cases is associated with the risk of being infected in communities. We

**Table 1 Characteristics of asymptomatic positive individuals.**

|  | Total (%) | Asymptomatic positive persons (%) | Detection rate per 10,000 (95% CI) | P value |
|---|---|---|---|---|
| *Total* | 9,899,828 (100.0) | 300 (100.0) | 0.303 (0.270–0.339) |  |
| Sex |  |  |  |  |
| Male | 5,162,960 (52.2) | 132 (44.0) | 0.256 (0.214–0.303) | 0.005 |
| Female | 4,736,868 (47.8) | 168 (56.0) | 0.355 (0.303–0.413 |  |
| Age (years old) |  |  |  |  |
| ≤17 | 969,014 (9.8) | 12 (4.0) | 0.124 (0.064–0.216) | <0.001 |
| 18–44 | 4,448,230 (44.9) | 104 (34.7) | 0.234 (0.191–0.283) |  |
| 45–59 | 2,492,943 (25.2) | 96 (32.0) | 0.385 (0.312–0.470) |  |
| ≥60 | 1,989,641 (20.1) | 88 (29.3) | 0.442 (0.355–0.545) |  |
| Administrative Districts in Wuhan |  |  |  |  |
| Wuchang | 904,636 (9.1) | 73 (24.3) | 0.807 (0.633–1.015) | <0.001 |
| Qingshan | 414,312 (4.2) | 23 (7.7) | 0.555 (0.352–0.833) |  |
| Qiaokou | 583,440 (5.9) | 32 (10.7) | 0.548 (0.375–0.774) |  |
| Hanyang | 717,429 (7.2) | 29 (9.7) | 0.404 (0.271–0.581) |  |
| Jianghan | 524,224 (5.3) | 19 (6.3) | 0.362 (0.218–0.566) |  |
| Hongshan | 1,103,079 (11.1) | 35 (11.7) | 0.317 (0.221–0.441) |  |
| East Lake High-tech Development Area | 782,987 (7.9) | 19 (6.3) | 0.243 (0.146–0.379) |  |
| Jiangan | 800,440 (8.1) | 19 (6.3) | 0.237 (0.143–0.371) |  |
| Caidian | 503,595 (5.1) | 11 (3.7) | 0.218 (0.109–0.391) |  |
| Jiangxia | 671,248 (6.8) | 14 (4.7) | 0.209 (0.114–0.350) |  |
| Huangpi | 979,920 (9.9) | 14 (4.7) | 0.143 (0.078–0.240) |  |
| Hannan | 417,022 (4.2) | 4 (1.3) | 0.096 (0.026–0.246) |  |
| Dongxihu | 777,204 (7.9) | 5 (1.7) | 0.064 (0.021–0.150) |  |
| Xinzhou | 634,408 (6.4) | 3 (1.0) | 0.047 (0.010–0.138) |  |
| East Lake Scenic Area of Wuhan | 85,884 (0.9) | 0 (0.0) | 0.000 (0.000–0.430) |  |

$\chi^2$ test was used to assess the association between the detection rate of asymptomatic cases increased and sex and age. Urban districts of Wuhan includes Wuchang, Qingshan, Qiaokou, Hanyang, Jiangan, Jianghan, and Hongshan; Suburban districts of Wuhan includes Hannan, Caidian, Dongxihu, Xinzhou, Jiangxia, Huangpi, East Lake High-tech Development Area, and East Lake Scenic Area of Wuhan.

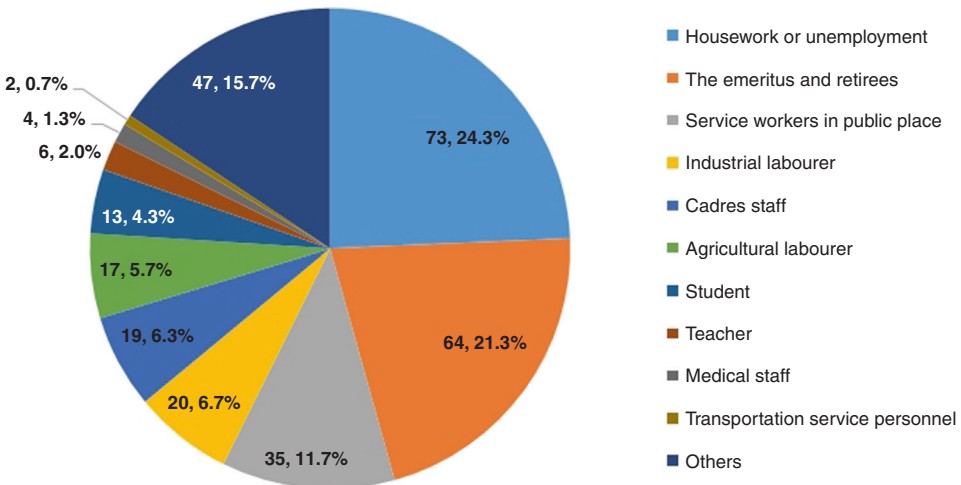

**Fig. 1 The occupation distribution of asymptomatic positive cases (%).** Note: Others included the self-employed, military personnel, and so on. (Source data are provided as s Source Data file.).

found that asymptomatic positive rates in different districts of Wuhan were correlated with the prevalence of previously confirmed cases. This is in line with the temporal and spatial evolution (especially the long-tailed characteristic) of infectious diseases[11].

Existing laboratory virus culture and genetic studies[9,10] showed that the virulence of SARS-CoV-2 virus may be weakening over time, and the newly infected persons were more likely to be asymptomatic and with a lower viral load than earlier infected cases. With the centralized isolation and treatment of all COVID-19 cases during the lockdown period in Wuhan, the risk of residents being infected in the community has been greatly reduced. When susceptible residents are exposed to a low dose of virus, they may tend to be asymptomatic as a result of their own immunity. Serological antibody testing in the current study found that at least 63% of asymptomatic positive cases were actually infected with SARS-CoV-2 virus. Nonetheless, it is too early to be complacent, because of the existence of asymptomatic positive cases and high level of susceptibility in residents in Wuhan. Public health measures for the prevention and control of COVID-19 epidemic, including wearing masks, keeping safe social distancing in Wuhan should be sustained. Especially, vulnerable populations with weakened immunity or co-morbidities, or both, should continue to be appropriately shielded.

Findings from this study show that COVID-19 was well controlled in Wuhan at the time of the screening programme. After two months since the screening programme (by August 9, 2020), there were no newly confirmed COVID-19 cases in Wuhan.

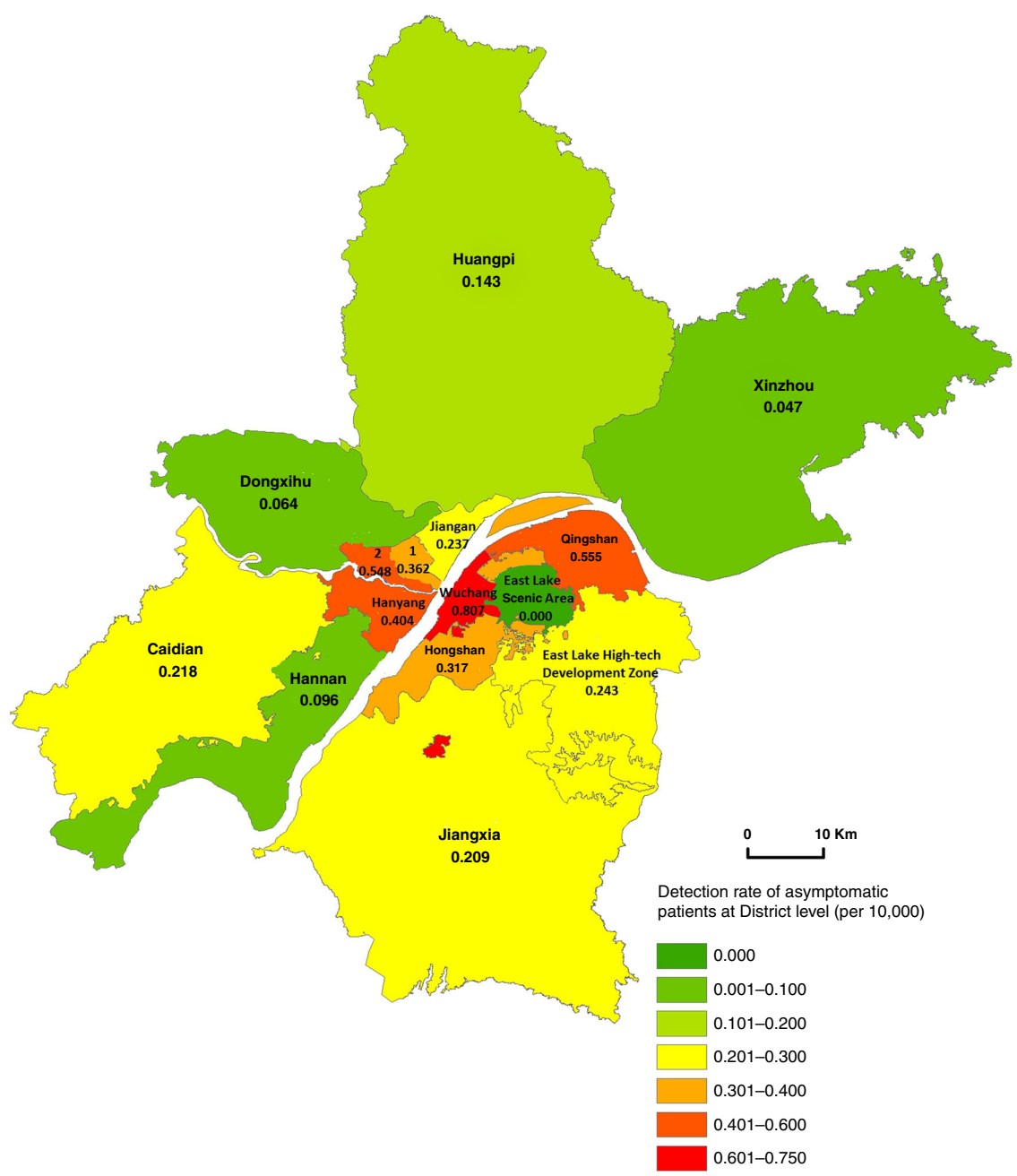

**Fig. 2 The geographic distribution of the detection rate of asymptomatic positive cases.** Note: 1 represents Jianghan district; 2 represents Qiaokou district. (Source data are provided as s Source Data file.).

**Table 2 Results of the detection of antibody in 300 asymptomatic positive persons.**

| IgM Results | IgG | Asymptomatic positive persons | % (95% CI) |
|---|---|---|---|
| − | + | 161 | 53.7 (47.8–59.4) |
| − | − | 110 | 36.7 (31.2–42.4) |
| + | + | 29 | 9.7 (6.6–13.6) |
| + | − | 0 | 0.0 (0.0–1.2) |

"−" indicates negative; "+" indicates positive.

Further testing of SARS-CoV-2 in samples collected from market environment settings in Wuhan were conducted, and found no positive results after checking a total of 52,312 samples from 1795 market setting during June 13 to July 2, 2020[12].

This study has several limitations that need to be discussed. First, this was a cross-sectional screening programme, and we are unable to assess the changes over time in asymptomatic positive and reoperative results. Second, although a positive result of nucleic acid testing reveals the existence of the viral RNAs, some false negative results were likely to have occurred, in particular due to the relatively low level of virus loads in asymptomatic infected individuals, inadequate collection of samples, and limited accuracy of the testing technology[13]. Although the screening programme provided no direct evidence on the sensitivity and specificity of the testing method used, a meta-analysis reported a

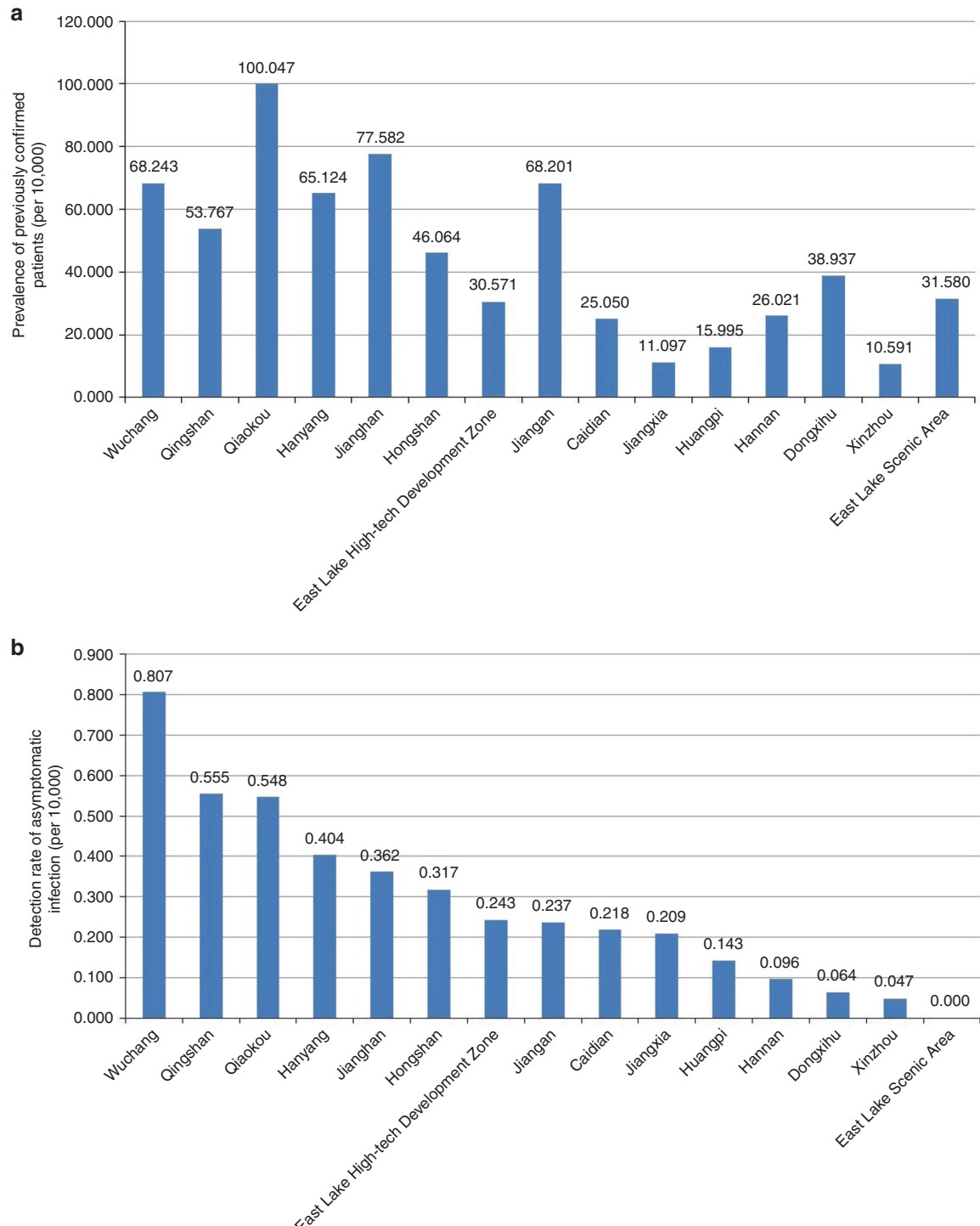

**Fig. 3 The prevalence of previously confirmed patients and the detection rate of asymptomatic positive cases of COVID-19 in each district in Wuhan.**
**a** The prevalence of previously confirmed patients of COVID-19 in each district in Wuhan. **b** The detection rate of asymptomatic positive cases of COVID-19 in each district in Wuhan. (Source data are provided as s Source Data file.).

pooled sensitivity of 73% (95% CI 68–78%) for nasopharayngeal and throat swab testing of COVID-19[14]. Testing kits used in the screening programme were publicly purchased by the government and these kits have been widely used in China and other countries. Multiple measures were taken to possibly minimise false negative results in the screening programme. For example, standard training was provided to health works for sample collection to ensure the sample quality. The experiment procedures, including specimen collection, extraction, PCR, were according to

official guidelines (Supplementary Note 1). For the real-time RT-PCR assay, two target genes were simultaneously tested. Even so, false negative results remained possible, particularly in any mass screening programmes. However, even if test sensitivity was as low as 50%, then the actual prevalence would be twice as high as reported in this study, but would still be very low. Around 7.1% of eligible residents did not participate in the citywide nucleic acid screening and the screening programme did not collect detailed data on reasons for nonparticipation, which is a limitation of this

study. Although there were no official statistics, a large number of migrant workers and university students left Wuhan before the lockdown, joining their families in other cities or provinces for traditional Chinese New Year. Therefore, it is likely that most nonparticipants were not in Wuhan at the time of the screening. The main objective of the screening programme was to assess the risk of COVID-19 epidemic in residents who were actually living in the post-lockdown Wuhan. Therefore, the estimated positive rates are unlikely to be materially influenced by nonparticipation of residents who were not in Wuhan or some residents who did not participate in the screening for other reasons. Moreover, people who left Wuhan were the target population for monitoring in other provinces and cities and were required to take nucleic acid testing. Although there was no official statistics showing the positive rate of nucleic acid testing in this population, there was no report that shown a higher positive rate of nucleic acid testing than our findings.

In summary, the detection rate of asymptomatic positive cases in the post-lockdown Wuhan was very low (0.303/10,000), and there was no evidence that the identified asymptomatic positive cases were infectious. These findings enabled decision makers to adjust prevention and control strategies in the post-lockdown period. Further studies are required to fully evaluate the impacts and cost-effectiveness of the citywide screening of SARS-CoV-2 infections on population's health, health behaviours, economy, and society.

## Methods

**Study population and ethical approvals.** Wuhan has about 11 million residents in total, with seven urban and eight suburban districts. Residents are living in 7280 residential communities (or residential enclosures, "xiao-qu" in Chinese), and each residential community could be physically isolated from other communities for preventing transmission of COVID-19.

The screening programme recruited residents (including recovered COVID-19 patients) currently living in Wuhan who were aged ≥6 years (5,162,960 males, 52.2%). All participants provided written or verbal informed consent after reading a statement that explained the purpose of the testing. For participants who aged 6–17 years old, consent was obtained from their parents or guardians. The study protocol for an evaluation of the programme based on anonymized screening data was approved by the Ethics Committee of the Tongji Medical College Institutional Review Board, Huazhong University of Science and Technology, Wuhan, China (No. IROG0003571).

**Organizational guarantee and community mobilization.** A citywide nucleic acid screening group was formed, with specialized task teams contributing to comprehensive coordination, technical guidance, quality control, participation invitation, information management, communication, and supervision of the screening. The city government invested 900 million yuan (RMB) in the testing programme. From 14 May to 1 June 2020, in the peak time, up to 2907 sample collection sites were functioning at the same time in Wuhan. Each sample collection site had an assigned sample collection group, including several health professionals (staffed according to the number of communities' residents), 2–4 community managers, 1–2 police officers, and 1–2 inspectors. The sampling sites were set up based on the number and accessibility of local residents. Local community workers were responsible for a safe and orderly sampling process to minimise the waiting time. In addition, mobile sampling teams were formed by primary health care professionals and volunteers to conduct door-to-door sampling for residents who had physical difficulties or were unable to walk.

About 50,000 health professionals (mainly doctors and nurses from community health centers) and more than 280,000 person-times of community workers and volunteers contributed to sample collection, transport of equipment and samples collected, arrangement of participation process, and maintaining order of sampling sites. Public information communication and participant invitation were implemented through mass media, mobile messages, WeChat groups, and residential community broadcasts, so as to increase residents' awareness and the participation.

**Acquisition, preservation, and transport of samples.** All sampling personnel received standard training for the collection of oropharyngeal swab samples. To minimise the risk of cross-infection, the sampling process strictly followed a disinfection process and environmental ventilation were ensured. The collected samples were stored in a virus preservation solution or immersed in isotonic saline, tissue culture solution, or phosphate buffer (Supplementary note 1). Then, all samples were sent to testing institutions within 4 h using delivery boxes for

biological samples refrigerated with dry ice to guarantee the stability of nucleic acid samples.

**Technical methods for laboratory testing of collected samples.** A total of 63 nucleic acid testing laboratories, 1451 laboratory workers and 701 testing equipment were involved in the nucleic acid testing. Received samples were stored at 4 °C and tested within 24 h of collection. Any samples that could not be tested within 24 h were stored at −70 °C or below (Supplementary note 1). In addition to "single testing" (i.e., separate testing of a single sample), "mixed testing" was also performed for 23% of the collected samples to increase efficiency, in which five samples were mixed in equal amounts, and tested in the same test tube. If a mixed testing was positive for COVID-19, all individual samples were separately retested within 24 h[15].

Details regarding technical methods for sequencing and virus culture were provided in Supplementary note 1. Real-time reverse transcriptase-polymerase chain reaction (RT-PCR) assay method was used for the nucleic acid testing. We simultaneously amplified and tested the two target genes: open reading frame 1ab (ORF1ab) and nucleocapsid protein (N) (Supplementary Note 1). A cycle threshold value (Ct-value) less than 37 was defined as a positive result, and no Ct-value or a Ct-value of 40 or more was defined as a negative result. For Ct-values ranging from 37 to 40, the sample was retested. If the retest result remained less than 40 and the amplification curve had obvious peak, the sample was classified as positive; otherwise, it was reported as being negative. These diagnostic criteria were based on China's official recommendations[16].

For asymptomatic positive cases, virus culture was carried out in biosafety level-3 laboratories. The colloidal gold antibody test was also performed for asymptomatic positive cases (Supplementary note 1). All testing results were double entered into a specifically designed database, and managed by the Big Data and Investigation Group of the COVID-19 Prevention and Control Centre in Wuhan, which was established to collect and manage data relevant to the COVID-19 epidemic.

**Participant data collection and management.** Before sample collection, residents electronically (using a specifically designed smartphone application) self-uploaded their personal information, including ID number, name, sex, age, and place of residence. Then, the electronic machine system generated a unique personal bar-code and stuck it on the sample tube to ensure the match between the sample and the participant. Then trained staff interviewed each individual regarding the history of COVID-19 and previous nucleic acid testing. There was a database of confirmed COVID-19 cases in Wuhan, which can be used to validate the self-reported previous COVID-19 infection. All information was entered into a central database. The testing results were continually uploaded to the central database by testing institutions. Contact tracing investigations were conducted on participants who tested positive for SARS-CoV-2, to track and manage their close contacts. The pre-existing unique identification code for each resident was used as the programme's identification number, to ensure information accuracy during the whole process of screening, from sampling, nucleic acid testing, result reporting, the isolation of detected positive cases, and tracing of close contacts of positive cases. All screening information was kept strictly confidential and was not allowed to be disclosed or used for other purposes other than clinical and public health management. Personal information of asymptomatic positive cases was only disclosed to designated medical institutions and community health centres for the purpose of medical isolation and identification of close contacts. Researcher was blind to the study hypothesis during data collection.

**Biological security guarantee.** Nucleic acid testing was performed in biosafety level-2 (BSL-2) laboratories, and virus culture was conducted in biosafety level-3 laboratories. Sampling and testing personnel adopted the personal protective measures according to the standard of biosafety level-3 laboratories. Participating laboratories implemented control measures to guarantee biological safety in accordance with relevant regulations[17].

**Result query and feedback.** Two to three days after sample collection, participants could inquire about their test results using WeChat or Alipay application by their unique ID numbers. The results included text descriptions of nucleic acid testing and coloured health codes. A green coloured health code refers to a negative result, and a red coloured health code indicates a positive result.

**Definition and management of identified confirmed cases and close contacts.** In this study, all confirmed COVID-19 cases were diagnosed by designated medical institutions according to National Guidelines for the Prevention and Control of COVID-19 (Supplementary Note 2). Asymptomatic positive cases referred to individuals who had a positive result during screening, and they had neither a history of COVID-19 diagnosis, nor any clinical symptoms at the time of the nucleic acid testing. Close contacts were individuals who closely contacted with an asymptomatic positive person since 2 days before the nucleic acid sampling[16]. Repositive cases refer to individuals who recovered from previously confirmed COVID-19 disease and had a positive testing again in the screening programme. All repositive cases, asymptomatic positive persons, and their close contacts were

isolated for at least 2 weeks in designated hotels managed by primary health care professionals, and they were released from isolation only if two consecutive nucleic acid tests were negative.

**Statistical analysis**. Detection rate of asymptomatic positive or repositive cases was calculated by dividing the number of individuals with a positive result of nucleic acid testing by the number of participants tested. Because of extremely low detection rates, we calculated 95% confidence intervals of estimated proportions using Pearson–Klopper exact method, implemented through R package "binom" version 1.1-1[18]. SPSS version 22.0 was used for other statistical analyses. We analyzed the distribution of asymptomatic positive cases and assessed the Spearman correlation between the asymptomatic positive rate and the prevalence of previously confirmed COVID-19 cases in different districts of Wuhan. Differences in asymptomatic positive rates by sex and age groups were assessed using the $\chi^2$ test. ArcGIS 10.0 was used to draw a geographic distribution map of asymptomatic positive cases. A value of $P < 0.05$ (two-tailed) was considered statistically significant.

**Reporting summary**. Further information on research design is available in the Nature Research Reporting Summary linked to this article.

## Data availability
Detailed data directly used to generate each figure or table of this study are available within the article, Supplementary Information and source data are provided with this paper.

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

## Acknowledgements
We would like to thank all institutions and all citizens in Wuhan for their support for citywide nucleic acid screening work. We also would like to thank the Wuhan city government for this citywide nucleic acid testing, sampling and management, and thank the big data and investigation group of COVID-19 prevention and control institution in Wuhan (the data and investigation group of Wuhan Municipal Health Commission) for their efforts in the data collection. In addition, we would like to thank the National Social Science Foundation of China (Grant No. 18ZDA085) for supporting the fund.

## Author contributions
S.Y.C., C.W., X.X.Y., and Z.X.L. conceived the study. C.W., Y.C.H., T.T.W., K.L., H.B.X., and X.S. participated in the acquisition of data. S.B.W. and J.G. were responsible for the on-site specimen collection, laboratory testing quality evaluation, and control. Y.C.H., T.T.W., and L.Q.L. analyzed the data. H.J., Y.H.G., and F.J.S. gave advice on methodology. Q.F.T. and C.Z.L. investigated the responses to the citywide nucleic acid testing among residents lived in outside of Wuhan city. S.Y.C., Y.G., C.W., and X.X.Y. drafted the manuscript, Y.G., M.B., and F.J.S. revised the manuscript, and M.B., C.Z.L., and F.J.S. critically commented and edited the manuscript. All authors read and approved the final manuscript. Z.X.L. is the guarantor of this study.

## Competing interests
The authors declare no competing interests.
