## [Peer Review file · Nature Communications]

Editorial note: This manuscript has been previously reviewed at another journal that is not operating a transparent peer review scheme. This document only contains reviewer comments and rebuttal letters for versions considered at *Nature Communications*.

REVIEWERS' COMMENTS

Reviewer #1 (Remarks to the Author):

The results of this paper are insightful and novel in their endeavor to determine potential cryptic infections and transmission that may be ongoing in the community. The authors have addressed their limitations and the queries appropriately and adequately to support their conclusion. It would be good if there is a separate seroprevalence study to further strengthen the extent of transmission among those swab negatives, if blood samples are available.

Reviewer #2 (Remarks to the Author):

I suggest that the last sentence of the abstract "promote the recovery of economy and normal social life in Wuhan" be deleted or rewritten.

A point-by-point response to the reviewers' comments, reproduced verbatim

Reviewer #1 (Remarks to the Author):

The results of this paper are insightful and novel in their endeavor to determine potential cryptic infections and transmission that may be ongoing in the community. The authors have addressed their limitations and the queries appropriately and adequately to support their conclusion. It would be good if there is a separate seroprevalence study to further strengthen the extent of transmission among those swab negatives, if blood samples are available.

Author's Response: We appreciate this good suggestion and agree with the reviewer.

However, the screening programme did not collect blood samples from individuals whose swab tests were negative. This important issue of seroprevalence among individuals with negative swab tests needs to be addressed in further studies.

Reviewer #2 (Remarks to the Author):

I suggest that the last sentence of the abstract "promote the recovery of economy and normal social life in Wuhan" be deleted or rewritten.

Author's Response: Thanks for the reviewer's suggestion. We have now deleted the last sentence of the abstract "promote the recovery of economy and normal social life in Wuhan".